# Efficiently Estimating Erdős-Rényi Graphs with Node Differential Privacy

**Adam Sealfon**
MIT and UC Berkeley
asealfon@berkeley.edu

**Jonathan Ullman**
Northeastern University
jullman@ccs.neu.edu

## Abstract

We give a simple, computationally efficient, and node-differentially-private algorithm for estimating the parameter of an Erdős-Rényi graph—that is, estimating $p$ in a $G(n, p)$—with near-optimal accuracy. Our algorithm nearly matches the information-theoretically optimal exponential-time algorithm for the same problem due to Borgs et al. (FOCS 2018). More generally, we give an optimal, computationally efficient, private algorithm for estimating the edge-density of any graph whose degree distribution is concentrated in a small interval.

## 1 Introduction

Network data modeling individuals and relationships between individuals are increasingly central in data science. As some of the most interesting network datasets include sensitive information about individuals, there is a need for private methods for analysis of these datasets, ideally satisfying strong mathematical guarantees like *differential privacy* [9]. However, while there is a highly successful literature on differentially private statistical estimation for traditional i.i.d. data, the literature on estimating network statistics is far less developed.

Early work on private network data focused on *edge differential privacy*, in which the algorithm is required to "hide" the presence or absence of a single edge in the graph (e.g. [20, 14, 16, 13, 1, 22, 17] and many more). A more desirable notion of privacy, which is the focus of this work, is *node differential privacy (node-DP)*, which requires the algorithm to hide the presence or absence of a single node and the (arbitrary) set of edges incident to that node.

However, node-DP is often difficult to achieve without compromising accuracy, because even very simple graph statistics can be highly sensitive to adding or removing a single node. For example, the count of edges in the graph, $|E|$, can change by $\pm n$ by adding or deleting a single node from an $n$-node graph, which means that no node-DP algorithm can count the number of edges with error $o(n)$ on a *worst-case* graph. We emphasize that even these simple statistics like the edge count can disclose sensitive information if no steps are taken to ensure privacy, especially when we release many such statistics on related graphs. There has been an enormous body of work that has uncovered the privacy risks of releasing simple statistics like counts in the i.i.d. setting (e.g. [8, 10, 12, 15, 19, 5, 11]) and the additional graph structure only makes these risks more acute.

Although node-DP is difficult to achieve on worst-case graphs, the beautiful works of Blocki et al. [2] and Kasiviswanathan et al. [18] showed how to design node-DP estimators that are highly accurate on "nice" graphs that have additional properties observed in practice—for example, graphs with small maximum degree—using the technique of Lipschitz extensions. However, many of the known constructions of Lipschitz extensions require exponential running time, and constructions of computationally efficient Lipschitz extensions [21, 7, 6] lag behind. As a result, even for estimating very simple graph models, there are large gaps in accuracy between the best known computationally efficient algorithms and the information theoretically optimal algorithms.

In this work we focus on arguably the simplest graph statistic, the *edge count*, $|E|$, in undirected unweighted graphs. We give improved estimators for this quantity on *concentrated-degree graphs*. Intuitively, a concentrated-degree graph is one in which the degree of every node lies in some small (but not publicly known) range $[\bar{d}-k, \bar{d}+k]$, which generalizes the case of graphs with low maximum degree. We give a *simple*, *polynomial-time* node-DP algorithm with *optimal accuracy* for estimating the count of edges in concentrated-degree graphs. Our estimator is inspired by Lipschitz extensions, but avoids directly constructing an efficient Lipschitz extension, and thus our approach may be useful for computing other graph statistics in settings where efficient Lipschitz extensions are unknown or unachievable.

The main application of this estimator is to estimate the parameter for the simplest possible network model, the *Erdős-Rényi graph*. In this model, denoted $G(n, p)$, we are given a number of nodes $n$ and a parameter $p \in [0, 1]$, and we sample an $n$-node graph $G$ by independently including each edge $(i, j)$ for $1 \le i < j \le n$ with probability $p$. The goal is to design a node-DP algorithm that takes as input a graph $G \sim G(n, p)$ and outputs an estimate $\hat{p} \approx p$. Surprisingly, until the elegant recent work of Borgs et al. [3], the optimal accuracy for estimating the parameter $p$ in a $G(n, p)$ via node-DP algorithms was unknown. Although that work essentially resolved the optimal accuracy of node-DP algorithms, their construction is again based on generic Lipschitz extensions, and thus results in an exponential-time algorithm, and, in our opinion, gives little insight for how to construct an efficient estimator with similar accuracy. Erdős-Rényi graphs automatically satisfy the concentrated-degree property with high probability, and thus we immediately obtain a computationally efficient, node-DP estimator for Erdős-Rényi graphs. The error of our estimator nearly matches that of Borgs et al., and indeed does match it for a wide range of parameters.

## 1.1 Background: Node-Private Algorithms for Erdős-Rényi Graphs

Without privacy, the optimal estimator is simply to output the *edge-density* $p_G = |E|/\binom{n}{2}$ of the realized graph $G \sim G(n, p)$, which guarantees that

$$\mathbb{E}_{G}\big[(p - p_G)^2\big] = \frac{p(1 - p)}{\binom{n}{2}}.$$

The simplest way to achieve $\varepsilon$-node-DP is to add zero-mean noise to the edge-density with standard-deviation calibrated to its *global-sensitivity*, which is the amount that changing the neighborhood of a single node in a graph can change its edge-density. The global sensitivity of $p_G$ is $\Theta(1/n)$, and thus the resulting private algorithm $\mathcal{A}_{\text{naïve}}$ satisfies

$$\mathbb{E}_{G}\big[(p - \mathcal{A}_{\text{naïve}}(G))^2\big] = \Theta(1/\varepsilon^2 n^2).$$

Note that this error is on the same order as or larger than the non-private error.

Borgs et al. [3] gave an improved $\varepsilon$-node-DP algorithm such that, when both $p$ and $\varepsilon$ are $\gtrsim \frac{\log n}{n}$,

$$\mathbb{E}\big[(p - \mathcal{A}_{\text{bcsz}}(G))^2\big] = \underbrace{\frac{p(1 - p)}{\binom{n}{2}}}_{\text{non-private error}} + \underbrace{\tilde{O}\Big(\frac{p}{\varepsilon^2 n^3}\Big)}_{\text{overhead due to privacy}}$$

What is remarkable about their algorithm is that, unless $\varepsilon$ is quite small (roughly $\varepsilon \lesssim n^{-1/2}$), the first term dominates the error, in which case privacy comes essentially *for free*. That is, the error of the private algorithm is only larger than that of the optimal non-private algorithm by a $1 + o(1)$ factor. However, as we discussed above, this algorithm is not computationally efficient.

The only computationally efficient node-DP algorithms for computing the edge-density apply to graphs with small maximum degree [2, 18, 21], and thus do not give optimal estimators for Erdős-Rényi graphs unless $p$ is very small.

## 1.2 Our Results

Our main result is a computationally efficient estimator for Erdős-Rényi graphs.

**Theorem 1.1** (Erdős-Rényi Graphs, Informal). *There is an $O(n^2)$-time $\varepsilon$-node-DP algorithm $\mathcal{A}$ such that for every $n$ and every $p \gtrsim 1/n$, if $G \sim G(n,p)$, then*

$$\mathop{\mathbb{E}}_{G,A}\big[(p - \mathcal{A}(G))^2\big] = \underbrace{\frac{p(1-p)}{\binom{n}{2}}}_{\textit{non-private error}} + \underbrace{\tilde{O}\left(\frac{p}{\varepsilon^2 n^3} + \frac{1}{\varepsilon^4 n^4}\right)}_{\textit{overhead due to privacy}}$$

The error of Theorem 1.1 matches that of the exponential-time estimator of Borgs et al. [3] up to the additive $\tilde{O}(1/\varepsilon^4 n^4)$ term, which is often not the dominant term in the overall error. In particular, the error of our estimator is still within a $1 + o(1)$ factor of the optimal non-private error unless $\varepsilon$ or $p$ is quite small—for example, when $p$ is a constant and $\varepsilon \gtrsim n^{-1/2}$.

Our estimator actually approximates the edge density for a significantly more general class of graphs than merely Erdős-Rényi graphs. Specifically, Theorem 1.1 follows from a more general result for the family of *concentrated-degree graphs*. For $k \in \mathbb{N}$, define $\mathcal{G}_{n,k}$ to be the set of $n$-node graphs such that the degree of every node is between $\bar{d} - k$ and $\bar{d} + k$, where $\bar{d} = 2|E|/n$ is the average degree of the graph.

**Theorem 1.2** (Concentrated-Degree Graphs, Informal). *For every $k \in \mathbb{N}$, there is an $O(n^2)$-time $\varepsilon$-node-DP algorithm $\mathcal{A}$ such that for every $n$ and every $G \in \mathcal{G}_{n,k}$,*

$$\mathop{\mathbb{E}}_{\mathcal{A}}\big[(p_G - \mathcal{A}(G))^2\big] = O\left(\frac{k^2}{\varepsilon^2 n^4} + \frac{1}{\varepsilon^4 n^4}\right)$$

*where $p_G = |E|/\binom{n}{2}$ is the empirical edge density of $G$.*

Theorem 1.1 follows from Theorem 1.2 by using the fact that for an Erdős-Rényi graph, with overwhelming probability the degree of every node lies in an interval of width $\tilde{O}(\sqrt{pn})$ around the average degree.

The main technical ingredient in Theorem 1.2 is to construct a *low sensitivity* estimator $f(G)$ for the number of edges. The first property we need is that when $G$ satisfies the concentrated degree property, $f(G)$ equals the number of edges in $G$. The second property of the estimator we construct is that its *smooth sensitivity* [20] is low on these graphs $G$. At a high level, the smooth sensitivity of $f$ at a graph $G$ is the most that changing the neighborhood of a small number of nodes in $G$ can change the value of $f(G)$. Once we have this property, it is sufficient to add noise to $f(G)$ calibrated to its smooth sensitivity. We construct $f$ by carefully reweighting edges that are incident on nodes that do not satisfy the concentrated-degree condition.

Finally, we are able to show that Theorem 1.2 is optimal for concentrated-degree graphs. In additional to being a natural class of graphs in its own right, this lower bound demonstrates that in order to improve Theorem 1.1, we will need techniques that are more specialized to Erdős-Rényi graphs.

**Theorem 1.3** (Lower Bound, Informal). *For every $n$ and $k$, and every $\varepsilon$-node-DP algorithm $A$, there is some $G \in \mathcal{G}_{n,k}$ such that $\mathop{\mathbb{E}}_{A}\big[(p_G - \mathcal{A}(G))^2\big] = \Omega\left(\frac{k^2}{\varepsilon^2 n^4} + \frac{1}{\varepsilon^4 n^4}\right)$. The same bound applies to $(\varepsilon, \delta)$-node-DP algorithms with sufficiently small $\delta \lesssim \varepsilon$.*

## 2 Preliminaries

Let $\mathcal{G}_n$ be the set of $n$-node graphs. We say that two graphs $G, G' \in \mathcal{G}_n$ are *node-adjacent*, denoted $G \sim G'$, if $G'$ can be obtained by $G$ modifying the neighborhood of a single node $i$. That is, there exists a single node $i$ such that for every edge $e$ in the symmetric difference of $G$ and $G'$, $e$ is incident on $i$. As is standard in the literature on differential privacy, we treat $n$ as a fixed quantity and define adjacency only for graphs with the same number of nodes. We could easily extend our definition of adjacency to include adding or deleting a single node itself.

**Definition 2.1** (Differential Privacy [9]). *A randomized algorithm $\mathcal{A}\colon \mathcal{G}_n \to \mathcal{R}$ is $(\varepsilon, \delta)$-node-differentially private if for every $G \sim G' \in \mathcal{G}_n$ and every $R \subseteq \mathcal{R}$,*

$$\mathbb{P}[\mathcal{A}(G) \in R] \leq e^\varepsilon \cdot \mathbb{P}[\mathcal{A}(G') \in R] + \delta.$$

*If $\delta = 0$ we will simply say that $\mathcal{A}$ is $\varepsilon$-node-differentially private. As we only consider node differential privacy in this work, we will frequently simply say that $A$ satisfies differential privacy.*

The next lemma is the basic composition property of differential privacy.

**Lemma 2.2** (Composition [9])**.** *If $\mathcal{A}_1, \mathcal{A}_2 : \mathcal{G}_n \to \mathcal{R}$ are each $(\varepsilon, \delta)$-node-differentially private algorithms, then the mechanism $\mathcal{A}(G) = (\mathcal{A}_1(G), \mathcal{A}_2(G))$ satisfies $(2\varepsilon, 2\delta)$-node-differential privacy. The same holds if $\mathcal{A}_2$ may depend on the output of $\mathcal{A}_1$.*

We will say that two graphs $G, G'$ are at *node distance* $c$ if there exists a sequence of graphs $G = G_0 \sim G_1 \sim \cdots \sim G_c = G'$. The standard *group privacy* property of differential privacy yields the following guarantees for graphs at node distance $c > 1$.

**Lemma 2.3** (Group Privacy [9])**.** *If $\mathcal{A} : \mathcal{G}_n \to \mathcal{R}$ is $(\varepsilon, \delta)$-node-differentially private and $G, G'$ are at node-distance $c$, then for every $R \subseteq \mathcal{R}$,*

$$\mathbb{P}[\mathcal{A}(G) \in R] \leq e^{c\varepsilon} \cdot \mathbb{P}[\mathcal{A}(G') \in R] + ce^{c\varepsilon}\delta.$$

**Sensitivity and Basic DP Mechanisms.** The main differentially private primitive we will use is *smooth sensitivity* [20]. Let $f : \mathcal{G}_n \to \mathbb{R}$ be a real-valued function. For a graph $G \in \mathcal{G}_n$, we can define the *local sensitivity* of $f$ at $G$ and the *global sensitivity* of $f$ to be

$$LS_f(G) = \max_{G' : G' \sim G} |f(G) - f(G')| \quad \text{and} \quad GS_f = \max_G LS_f(G) = \max_{G' \sim G} |f(G) - f(G')|.$$

A basic result in differential privacy says that we can achieve privacy for any real-valued function $f$ by adding noise calibrated to the global sensitivity of $f$.

**Theorem 2.4** (DP via Global Sensitivity [9])**.** *Let $f : \mathcal{G}_n \to \mathbb{R}$ be any function. Then the algorithm $\mathcal{A}(G) = f(G) + \frac{GS_f}{\varepsilon} \cdot Z$, where $Z$ is sampled from a standard Laplace distribution,[1] satisfies $(\varepsilon, 0)$-differential privacy. Moreover, this mechanism satisfies $\underset{\mathcal{A}}{\mathbb{E}}\big[(\mathcal{A}(G) - f(G))^2\big] = O(GS_f/\varepsilon)$, and for every $t > 0$, $\underset{\mathcal{A}}{\mathbb{P}}[|\mathcal{A}(G) - f(G)| \geq t \cdot GS_f/\varepsilon] \leq \exp(-t)$.*

In many cases the global sensitivity of $f$ is too high, and we want to use a more refined mechanism that adds instance-dependent noise that is more comparable to the local sensitivity. This can be achieved via the *smooth sensitivity* framework of Nissim et al. [20].

**Definition 2.5** (Smooth Upper Bound [20])**.** *Let $f : \mathcal{G}_n \to \mathbb{R}$ be a real-valued function and $\beta > 0$ be a parameter. A function $S : \mathcal{G}_n \to \mathbb{R}$ is a $\beta$-smooth upper bound on $LS_f$ if*

    *1. for all $G \in \mathcal{G}_n$, $S(G) \geq LS_f(G)$, and*

    *2. for all neighboring $G \sim G' \in \mathcal{G}_n$, $S(G) \leq e^{\beta} \cdot S(G')$.*

The key result in smooth sensitivity is that we can achieve differential privacy by adding noise to $f(G)$ proportional to any smooth upper bound $S(G)$.

**Theorem 2.6** (DP via Smooth Sensitivity [20, 4])**.** *Let $f : \mathcal{G}_n \to \mathbb{R}$ be any function and $S$ be a $\beta$-smooth upper bound on the local sensitivity of $f$ for any $\beta \leq \varepsilon$. Then the algorithm $\mathcal{A}(G) = f(G) + \frac{S(G)}{\varepsilon} \cdot Z$, where $Z$ is sampled from a Student's $t$-distribution with 3 degrees of freedom,[2] satisfies $(\tilde{O}(\varepsilon), 0)$-differential privacy.*

*Moreover, for any $G \in \mathcal{G}_n$, this algorithm satisfies $\underset{\mathcal{A}}{\mathbb{E}}\big[(\mathcal{A}(G) - f(G))^2\big] = O(S(G)^2/\varepsilon^2)$.*

## 3   An Estimator for Concentrated-Degree Graphs

### 3.1   The Estimator

In order to describe the estimator we introduce some key notation. The input to the estimator is a graph $G = (V, E)$ and a parameter $k^*$. Intuitively, $k^*$ should be an upper bound on the concentration

**Algorithm 1:** Estimating the edge density of a concentrated-degree graph.

---

**Input**: A graph $G \in \mathcal{G}_n$ and parameters $\varepsilon > 0$ and $k^* \geq 0$.
**Output**: A parameter $0 \leq \hat{p} \leq 1$.

Let $p_G = \frac{1}{\binom{n}{2}} \sum_e x_e$ and $\bar{d}_G = (n-1)p_G$.

Let $\beta = \min(\varepsilon, 1/\sqrt{k^*})$.

Let $k_G > 0$ be the smallest positive integer such that at most $k_G$ vertices have degree outside $[\bar{d}_G - k^* - 3k_G, \bar{d}_G + k^* + 3k_G]$.

For $v \in V$, let $t_v = \min\{|t| : \deg_G(v) \pm t \in [\bar{d}_G - k^* - 3k_G, \bar{d}_G + k^* + 3k_G]\}$ and let $\mathsf{wt}_G(v) = \max(0, 1 - \beta t_v)$.

For each $u, v \in V$, let $\mathsf{wt}_G(\{u,v\}) = \min(\mathsf{wt}_G(u), \mathsf{wt}_G(v))$ and let $\mathsf{val}_G(e) = \mathsf{wt}_G(e) \cdot x_e + (1 - \mathsf{wt}_G(e))p_G$.

Let $f(G) = \sum_{u \neq v} \mathsf{val}_G(\{u,v\})$, where the sum is over unordered pairs of vertices.

Let
$$s = \max_{\ell \in L} 210 \cdot e^{-\beta \ell} \cdot (k_G + \ell + k^* + \beta(k_G + \ell)(k_G + \ell + k^*) + 1/\beta),$$
where $L = \{0, \lfloor 1/\beta - k_G - k^* \rfloor, \lceil 1/\beta - k_G - k^* \rceil\}$.

Return $\frac{1}{\binom{n}{2}} \cdot (f(G) + (s/\varepsilon) \cdot Z)$, where $Z$ is sampled from a Student's $t$-distribution with three degrees of freedom.

---

parameter of the graph, although we obtain more general results when $k^*$ is not an upper bound, in case the user does not have an *a priori* upper bound on this quantity.

For a graph $G = (V, E)$, let $p_G = |E|/\binom{n}{2}$ be the empirical edge density of $G$, and let $\bar{d}_G = (n-1)p_G$ be the empirical average degree of $G$. Let $k_G$ be the smallest positive integer value such that at most $k_G$ vertices of $G$ have degree differing from $\bar{d}_G$ by more than $k'_G := k^* + 3k_G$. Define $I_G = [\bar{d}_G - k'_G, \bar{d}_G + k'_G]$. For each vertex $v \in V$, let $t_v = \min\{|t| : \deg_G(v) \pm t \in I_G\}$ be the distance between $\deg_G(v)$ and the interval $I_G$, and define the *weight* $\mathsf{wt}_G(v)$ of $v$ as follows. For a parameter $\beta > 0$ to be specified later, let

$$\mathsf{wt}_G(v) = \begin{cases} 1 & \text{if } t_v = 0 \\ 1 - \beta t_v & \text{if } t_v \in (0, 1/\beta] \\ 0 & \text{otherwise.} \end{cases}$$

That is, $\mathsf{wt}_G(v) = \max(0, 1 - \beta t_v)$. For each pair of vertices $e = \{u, v\}$, define the *weight* $\mathsf{wt}_G(e)$ and *value* $\mathsf{val}_G(e)$ as follows. Let

$$\mathsf{wt}_G(e) = \min(\mathsf{wt}_G(u), \mathsf{wt}_G(v)) \quad \text{and} \quad \mathsf{val}_G(e) = \mathsf{wt}_G(e) \cdot x_e + (1 - \mathsf{wt}_G(e)) \cdot p_G,$$

where $x_e$ denotes the indicator variable on whether $e \in E$. Define the function $f(G) = \sum_{u,v \in V} \mathsf{val}_G(\{u,v\})$ to be the total value of all pairs of vertices in the graph, where the sum is over unordered pairs of distinct vertices.

Once we construct this function $f$, we add noise to $f$ proportional to a $\beta$-smooth upper bound on the sensitivity of $f$, which we derive in this section. Pseudocode for our estimator is given in Algorithm 1.

## 3.2 Analysis Using Smooth Sensitivity

We begin by bounding the local sensitivity $LS_f(G)$ of the function $f$ defined above.

**Lemma 3.1.** *For $\beta = \Omega(1/n)$, we have that $LS_f(G) = O((k_G + k^*)(1 + \beta k_G) + \frac{1}{\beta})$. In particular, for $\beta \in [1/n, 1]$, we have $LS_f(G) < 210((k_G + k^*)(1 + \beta k_G) + 1/\beta)$.*

*Proof.* Consider any pair of graphs $G, G'$ differing in only a single vertex $v^*$, and note that the empirical edge densities $p_G$ and $p_{G'}$ can differ by at most $\frac{2}{n} < \frac{2}{n-1}$, so $\bar{d}_G$ and $\bar{d}_{G'}$ can differ by at most 2. Moreover, for any vertex $v \neq v^*$, the degree of $v$ can differ by at most 1 between $G$ and $G'$. Consequently, by the Triangle Inequality, for any $v \neq v^*$, $|\bar{d}_G - \deg_G(v)|$ can differ from $|\bar{d}_{G'} - \deg_{G'}(v)|$ by at most 3 and $|k_G - k_{G'}| \leq 1$, so $\mathsf{wt}_G(v)$ can differ from $\mathsf{wt}_{G'}(v)$ by at most $6\beta$.

Let $\mathsf{Far}_G$ denote the set of at most $k_G$ vertices whose degree differs from $\bar{d}_G$ by more than $k_G' = k^* + 3k_G$. For any vertices $u, v \notin \mathsf{Far}_G \cup \mathsf{Far}_{G'} \cup \{v^*\}$, we have $\mathsf{wt}_G(\{u, v\}) = \mathsf{wt}_{G'}(\{u, v\}) = 1$, so $\mathsf{val}_G(\{u, v\}) = \mathsf{val}_{G'}(\{u, v\})$, since the edge $\{u, v\}$ appears in $G$ if and only if it appears in $G'$.

Now consider edges $\{u, v\}$ such that $u, v \neq v^*$ but $u \in \mathsf{Far}_G \cup \mathsf{Far}_{G'}$ (and $v$ may or may not be as well). If $\deg_G(u) \notin [\bar{d}_G - k_G'', \bar{d}_G + k_G'']$ for $k_G'' = k_G' + 1/\beta + 3$, then $\mathsf{wt}_G(u) = \mathsf{wt}_{G'}(u) = 0$ and so $|\mathsf{val}_G(\{u, v\}) - \mathsf{val}_{G'}(\{u, v\})| = |p_G - p_{G'}| \leq 2/n$. Otherwise, $\deg_G(u) \in [\bar{d}_G - k_G'', \bar{d}_G + k_G'']$. We can break up the sum

$$f_u(G) := \sum_{v \neq u} \mathsf{val}_G(\{u, v\}) = \sum_{v \neq u} \mathsf{wt}_G(\{u, v\}) \cdot x_{\{u, v\}} + \sum_{v \neq u}(1 - \mathsf{wt}_G(\{u, v\}))p_G.$$

Since at most $k_G$ other vertices can have weight less than that of $u$, we can bound the first term by

$$\sum_{v \neq u} \mathsf{wt}_G(u)x_{\{u, v\}} \pm k_G\mathsf{wt}_G(u) = \deg_G(u)\mathsf{wt}_G(u) \pm k_G\mathsf{wt}_G(u)$$

and the second term by

$$p_G \cdot \left((n-1) - \sum_{v \neq u} \mathsf{wt}_G(\{u, v\})\right) = \bar{d}_G - \bar{d}_G\mathsf{wt}_G(u) \pm p_Gk_G\mathsf{wt}_G(u)$$

so the total sum is bounded by $f_u(G) = \bar{d}_G + (\deg_G(u) - \bar{d}_G)\mathsf{wt}_G(u) \pm 2k_G\mathsf{wt}_G(u)$. Since $|\mathsf{wt}_G(u) - \mathsf{wt}_{G'}(u)| \leq 6\beta$, it follows that

$$|f_u(G) - f_u(G')| \leq 7 + 6\beta(k_G'' + 3) + 9\beta + 6\beta k_G$$
$$= 13 + 45\beta + 6\beta(k^* + 4k_G)$$
$$= O(1 + \beta(k_G + k^*)).$$

Since there are at most $k_G + k_G' \leq 2k_G + 1$ vertices in $u \in \mathsf{Far}_G \cup \mathsf{Far}_{G'} \setminus \{v^*\}$, the total difference in the terms of $f(G)$ and $f(G')$ corresponding to such vertices is at most $2k_G + 1$ times this, which is $O(k_G + \beta k_G(k_G + k^*))$. However, we are double-counting any edges between two vertices in $u \in \mathsf{Far}_G \cup \mathsf{Far}_{G'}$; the number of such edges is at most $2k_G^2 + k_G = O(k_G^2)$, and for any such edge $e$, $|\mathsf{val}_G(e) - \mathsf{val}_{G'}(e)| \leq 12\beta + 2/n = O(\beta + 1/n)$. Consequently the error induced by this double-counting is at most $(2k_G^2 + k_G)(12\beta + 2/n)$, which is $O(\beta k_G^2 + k_G^2/n)$, so the total difference between the terms of $f(G)$ and $f(G')$ corresponding to such vertices is at most

$$13 + 26k_G + 45\beta + 126\beta k_G + 6\beta k^* + 12\beta k^* k_G + 72\beta k_G^2 + 6k_G^2/n,$$

which is still $O(k_G + \beta k_G(k_G + k^*))$ for $\beta = \Omega(1/n)$.

Finally, consider the edges $\{u, v^*\}$ involving vertex $v^*$. If $\mathsf{wt}_G(v^*) = 0$ then

$$f_{v^*}(G) = \sum_{v \neq v^*} \mathsf{val}_G(\{v^*, v\}) = (n-1)p_G = \bar{d}_G.$$

If $\mathsf{wt}_G(v^*) = 1$ then $\deg_G(v^*) \in [\bar{d}_G - k_G', \bar{d}_G + k_G']$, so

$$f_{v^*}(G) = \sum_{v \neq v^*} \mathsf{val}_G(\{v^*, v\}) = \deg_G(v^*) \pm k_G = \bar{d}_G \pm k_G' \pm k_G.$$

Otherwise, $\deg_G(v^*) \in [\bar{d}_G - k_G' - 1/\beta, \bar{d}_G + k_G' + 1/\beta]$. Then we have that

$$f_{v^*}(G) = \sum_{v \neq v^*} \mathsf{val}_G(\{v^*, v\})$$
$$= \bar{d}_G + (\deg_G(v^*) - \bar{d}_G)\mathsf{wt}_G(v^*) \pm k_G\mathsf{wt}_G(v^*)$$
$$= \bar{d}_G \pm (\deg_G(v^*) - \bar{d}_G) \pm k_G,$$

so in either case we have that $f_{v^*}(G) \in [\bar{d}_G - (k'_G + k_G + 1/\beta), \bar{d}_G + (k'_G + k_G + 1/\beta)]$. Consequently $|f_{v^*}(G) - f_{v^*}(G')| \leq 3 + 8k_G + 2k^* + 2/\beta = O(k_G + k^* + 1/\beta)$.

Putting everything together, we have that

$$LS_f(G) \leq 16 + 34k_G + 2k^* + 45\beta + 126\beta k_G + 6\beta k^* + 12\beta k^* k_G + 72\beta k_G^2 + 6k_G^2/n + 2/\beta,$$

which is $O((k_G + k^*)(1 + \beta k_G) + 1/\beta)$ for $\beta = \Omega(1/n)$. In particular, for $\beta \in [1/n, 1]$, we have that $LS_f(G) \leq 210((k_G + k^*)(1 + \beta k_G) + \frac{1}{\beta})$. $\qquad\square$

We now compute a smooth upper bound on $LS_f(G)$. Let

$$g(k_G, k^*, \beta) = 210((k_G + k^*)(1 + \beta k_G) + \frac{1}{\beta})$$

be the upper bound on $LS_f(G)$ from Lemma 3.1, and let

$$S(G) = \max_{\ell \geq 0} e^{-\ell\beta} g(k_G + \ell, k^*, \beta).$$

**Lemma 3.2.** $S(G)$ is a $\beta$-smooth upper bound on the local sensitivity of $f$. Moreover, we have the bound $S(G) = O((k_G + k^*)(1 + \beta k_G) + \frac{1}{\beta})$.

*Proof.* For neighboring graphs $G, G'$, we have that

$$
\begin{aligned}
S(G') &= \max_{\ell \geq 0} e^{-\ell\beta} g(k_{G'} + \ell, k^*, \beta) \\
&\leq \max_{\ell \geq 0} e^{-\ell\beta} g(k_G + \ell + 1, k^*, \beta) \\
&= e^\beta \max_{\ell \geq 1} e^{-\ell\beta} g(k_G + \ell, k^*, \beta) \\
&\leq e^\beta \max_{\ell \geq 0} e^{-\ell\beta} g(k_G + \ell, k^*, \beta) \\
&= e^\beta S(G).
\end{aligned}
$$

Moreover, for fixed $k_G, k^*, \beta$, consider the function $h(\ell) = e^{-\ell\beta} g(k_G + \ell, k^*, \beta)$, and consider the derivative $h'(\ell)$. We have that $h'(\ell) = 210 \cdot \beta e^{-\ell\beta}(k_G + \ell)(1 - \beta(k_G + \ell + k^*))$. Consequently the only possible local maximum for $\ell > 0$ would occur for $\ell = 1/\beta - k_G - k^*$; note that the function $h$ decreases as $\ell \to \infty$. Consequently the maximum value of $h$ occurs for some $\ell \leq 1/\beta$, and so we can show by calculation that $S(G) < 630 \cdot ((k_G + k^*)(1 + \beta k_G) + \frac{1}{\beta})$ as desired. $\qquad\square$

*Remark.* Note that $S(G)$ can be computed efficiently, since $\ell$ can be restricted to the nonnegative integers and so the only candidate values for $\ell$ are 0, $\lfloor 1/\beta - k_G - k^* \rfloor$, and $\lceil 1/\beta - k_G - k^* \rceil$.

**Theorem 3.3.** *Algorithm 1 is $(O(\varepsilon), 0)$-differentially private for $\varepsilon \geq 1/n$. Moreover, for any $k$-concentrated $n$-vertex graph $G = (V, E)$ with $k \geq 1$, we have that Algorithm 1 satisfies*

$$\mathbb{E}_{\mathcal{A}}\left[\left(\frac{|E|}{\binom{n}{2}} - \mathcal{A}_{\varepsilon,k}(G)\right)^2\right] = O\left(\frac{k^2}{\varepsilon^2 n^4} + \frac{1}{\varepsilon^4 n^4}\right)$$

*Proof.* Algorithm 1 computes function $f$ and releases it with noise proportional to a $\beta$-smooth upper bound on the local sensitivity for $\beta \leq \varepsilon$. Consequently $(O(\varepsilon), 0)$-differential privacy follows immediately from Theorem 2.6.

We now analyze its accuracy on $k$-concentrated graphs $G$. If $G$ is $k$-concentrated and $k^* \geq k$, then $\mathsf{wt}_G(v) = 1$ for all vertices $v \in V$ and $\mathsf{val}_G(\{u, v\}) = x_{\{u,v\}}$ for all $u, v \in V$, and so $f(G) = |E|$. Consequently Algorithm 1 computes the edge density of a $k$-concentrated graph with noise distributed according to the Student's $t$-distribution scaled by a factor of $S(G)/(\varepsilon\binom{n}{2})$.

Since $G$ is $k$-concentrated, we also have that $k_G = 1$, and so $S(G) = O(k + \beta(k + 1) + 1/\beta) \leq O(k + 1/\varepsilon)$ by Lemma 3.2. The variance of the Student's $t$-distribution with three degrees of freedom is $O(1)$, so the expected squared error of the algorithm is

$$O\left(\frac{(k + 1/\varepsilon)^2}{\varepsilon^2 n^4}\right) = O\left(\frac{k^2}{\varepsilon^2 n^2} + \frac{1}{\varepsilon^4 n^4}\right)$$

as desired. $\qquad\square$

# 4 Application to Erdős-Rényi Graphs

In this section we show how to apply Algorithm 1 to estimate the parameter of an Erdős-Rényi graph.

---

**Algorithm 2:** Estimating the parameter of an Erdős-Rényi graph.

**Input**: A graph $G \in \mathcal{G}_n$ and parameters $\varepsilon, \alpha > 0$.
**Output**: A parameter $0 \leq \hat{p} \leq 1$.

Let $\tilde{p}' \leftarrow \frac{1}{\binom{n}{2}} \sum_e x_e + (2/\varepsilon n) \cdot Z$ where $Z$ is a standard Laplace

Let $\tilde{p} \leftarrow \tilde{p}' + 4 \log(1/\alpha)/\varepsilon n$ and $\tilde{k} \leftarrow \sqrt{\tilde{p} n \log(n/\alpha)}$

Return $\hat{p} \leftarrow \mathcal{A}_{\tilde{k},\varepsilon}(G)$ where $\mathcal{A}_{\tilde{k},\varepsilon}$ is Algorithm 1 with parameters $\tilde{k}$ and $\varepsilon$

---

It is straightforward to prove that this mechanism satisfies differential privacy.

**Theorem 4.1.** *Algorithm 2 satisfies* $(O(\varepsilon), 0)$-*node-differential privacy for* $\varepsilon \geq 1/n$.

*Proof.* The first line computes the empirical edge density of the graph $G$, which is a function with global sensitivity $(n-1)/\binom{n}{2} = 2/n$. Therefore by Theorem 2.4 this step satisfies $(\varepsilon, 0)$-differential privacy. The third line runs an algorithm that satisfies $(O(\varepsilon), 0)$-differential privacy for every fixed parameter $\tilde{k}$. By Lemma 2.2, the composition satisfies $(O(\varepsilon), 0)$-differential privacy. $\square$

Next, we argue that this algorithm satisfies the desired accuracy guarantee.

**Theorem 4.2.** *For every* $n \in \mathbb{N}$ *and* $\frac{1}{2} \geq p \geq 0$, *and an appropriate parameter* $\alpha > 0$, *Algorithm 2 satisfies*

$$\mathop{\mathbb{E}}_{G \sim G(n,p),\mathcal{A}} \left[ (p - \mathcal{A}(G))^2 \right] = \frac{p(1-p)}{\binom{n}{2}} + \tilde{O}\left( \frac{\max\{p, \frac{1}{n}\}}{\varepsilon^2 n^3} + \frac{1}{\varepsilon^4 n^4} \right)$$

*Proof.* We will prove the result in the case where $p \geq \frac{\log n}{n}$. The case where $p$ is smaller will follow immediately by using $\frac{\log n}{n}$ as an upper bound on $p$. The first term in the bound is simply the variance of the empirical edge-density $\bar{p}$. For the remainder of the proof we will focus on bounding $\mathbb{E}\left[ (\bar{p} - \hat{p})^2 \right]$.

A basic fact about $G(n,p)$ for $p \geq \frac{\log n}{n}$ is that with probability at least $1 - 2\alpha$: (1) $|\bar{p} - p| \leq 2\log(1/\alpha)/n$, and (2) the degree of every node $i$ lies in the interval $[\bar{d} \pm \sqrt{pn \log(n/\alpha)}]$ where $\bar{d}$ is the average degree of $G$. We will assume for the remainder that these events hold.

Using Theorem 2.4, we also have that with probability at least $1 - \alpha$, the estimate $\tilde{p}'$ satisfies $|\bar{p} - \tilde{p}'| \leq 4\log(1/\alpha)/\varepsilon n$. We will also assume for the remainder that this latter event holds. Therefore, we have $p \leq \tilde{p}$ and $p \geq \tilde{p} - 8\log(1/\alpha)/\varepsilon n$.

Assuming this condition holds, the graph will have $\tilde{k}$ concentrated degrees for $\tilde{k}$ as specified on line 2 of the algorithm. Since this assumption holds, we have

$$\mathbb{E}\left[ (\bar{p} - \mathcal{A}_{\tilde{k},\varepsilon}(G))^2 \right] = \tilde{O}\left( \frac{\tilde{k}^2}{\varepsilon^2 n^4} + \frac{1}{\varepsilon^4 n^4} \right) = \tilde{O}\left( \frac{pn + \frac{1}{\varepsilon n}}{\varepsilon^2 n^4} + \frac{1}{\varepsilon^4 n^4} \right) = \tilde{O}\left( \frac{pn}{\varepsilon^2 n^4} + \frac{1}{\varepsilon^4 n^4} \right)$$

To complete the proof, we can plug in a suitably small $\alpha = 1/\text{poly}(n)$ so that the $O(\alpha)$ probability of failure will not affect the overall mean-squared error in a significant way. $\square$

# 5 Lower Bounds for Concentrated-Degree Graphs

In this section we prove a lower bound for estimating the number of edges in concentrated-degree graphs. Theorem 5.1, which lower bounds the mean squared error, follows from Jensen's Inequality.

**Theorem 5.1.** *For every $n, k \in \mathbb{N}$, every $\varepsilon \in [\frac{2}{n}, \frac{1}{4}]$ and $\delta \le \frac{\varepsilon}{32}$, and every $(\varepsilon, \delta)$-node-DP algorithm $\mathcal{A}$, there exists $G \in \mathcal{G}_{n,k}$ such that $\mathbb{E}_{\mathcal{A}}[|p_G - \mathcal{A}(G)|] = \Omega\left(\frac{k}{\varepsilon n^2} + \frac{1}{\varepsilon^2 n^2}\right)$.*

The proof relies only on the following standard fact about differentially private algorithms.

**Lemma 5.2.** *Suppose there are two graphs $G_0, G_1 \in \mathcal{G}_{n,k}$ at node distance at most $\frac{1}{\varepsilon}$ from one another. Then for every $(\varepsilon, \frac{\varepsilon}{32})$-node-DP algorithm $\mathcal{A}$, there exists $b \in \{0, 1\}$ such that $\mathbb{E}_{\mathcal{A}}[|p_{G_b} - \mathcal{A}(G_b)|] = \Omega(|p_{G_0} - p_{G_1}|)$.*

We will construct two simple pairs of graphs to which we can apply Lemma 5.2.

**Lemma 5.3** (Lower bound for large $k$). *For every $n, k \in \mathbb{N}$ and $\varepsilon \ge 2/n$, there is a pair of graphs $G_0, G_1 \in \mathcal{G}_{n,k}$ at node distance $1/\varepsilon$ such that $|p_{G_0} - p_{G_1}| = \Omega(\frac{k}{\varepsilon n^2})$.*

*Proof.* Let $G_0$ be the empty graph on $n$ nodes. Note that $p_{G_0} = 0$, $\bar{d}_{G_0} = 0$, and $G_0$ is in $\mathcal{G}_{n,k}$.

We construct $G_1$ as follows. Start with the empty bipartite graph with $\frac{1}{\varepsilon}$ nodes on the left and $n - \frac{1}{\varepsilon}$ nodes on the right. We connect the first node on the left to each of the first $k$ nodes on the right, then the second node on the left to each of the next $k$ nodes on the right and so on, wrapping around to the first node on the right when we run out of nodes. By construction, $p_{G_1} = k/\varepsilon \binom{n}{2}$, $\bar{d}_{G_1} = 2k/\varepsilon n$. Moreover, each of the first $\frac{1}{\varepsilon}$ nodes has degree exactly $k$ and each of the nodes on the right has degree $\frac{k/\varepsilon}{n - 1/\varepsilon} \pm 1 = \frac{k}{\varepsilon n - 1} \pm 1$ Thus, for $n$ larger than some absolute constant, every degree lies in the interval $[\bar{d}_{G_1} \pm k]$ so we have $G_1 \in \mathcal{G}_{n,k}$. $\qquad\square$

**Lemma 5.4** (Lower bound for small $k$). *For every $n \ge 4$ and $\varepsilon \in [2/n, 1/4]$, there is a pair of graphs $G_0, G_1 \in \mathcal{G}_{n,1}$ at node distance $1/\varepsilon$ such that $|p_{G_0} - p_{G_1}| = \Omega(\frac{1}{\varepsilon^2 n^2})$.*

*Proof.* Let $i = \lceil n\varepsilon \rceil$, and let $G_0$ be the graph consisting of $i$ disjoint cliques each of size $\lfloor n/i \rfloor$ or $\lceil n/i \rceil$. Let $G_1$ be the graph consisting of $i+1$ disjoint cliques each of size $\lfloor n/(i+1) \rfloor$ or $\lceil n/(i+1) \rceil$. We can obtain $G_0$ from $G_1$ by taking one of the cliques and redistributing its vertices among the $i$ remaining cliques, so $G_0$ and $G_1$ have node distance $\ell := \lfloor n/(i+1) \rfloor \le 1/\varepsilon$. For $1/4 \ge \varepsilon \ge 2/n$ we have that $\ell \ge \lfloor 1/2\varepsilon \rfloor > 1/4\varepsilon$. Transforming $G_1$ into $G_0$ involves removing a clique of size $\ell$, containing $\binom{\ell}{2}$ edges, and then inserting these $\ell$ vertices into cliques that already have size $\ell$, adding at least $\ell^2$ new edges. Consequently $G_0$ contains at least $\ell^2 - \ell(\ell-1)/2 = \ell(\ell+1)/2$ more edges than $G_1$, so

$$|p_{G_1} - p_{G_0}| \ge \frac{\binom{\ell+1}{2}}{\binom{n}{2}} \ge \frac{\ell^2}{n^2} \ge \Omega(1/\varepsilon^2 n^2),$$

as desired. $\qquad\square$

Theorem 5.1 now follows by combining Lemmas 5.2, 5.3, and 5.4.

## Acknowledgments

Part of this work was done while the authors were visiting the Simons Institute for the Theory of Computing. AS is supported by NSF MACS CNS-1413920, DARPA/NJIT Palisade 491512803, Sloan/NJIT 996698, and MIT/IBM W1771646. JU is supported by NSF grants CCF-1718088, CCF-1750640, and CNS-1816028. The authors are grateful to Adam Smith for helpful discussions.

## Footnotes

[1]The standard Laplace distribution $Z$ has $\mathbb{E}[Z] = 0$, $\mathbb{E}\big[Z^2\big] = 2$, and density $\mu(z) \propto e^{-|z|}$.

[2]The Student's $t$-distribution with 3 degrees of freedom can be efficiently sampled by choosing $X, Y_1, Y_2, Y_3 \sim \mathcal{N}(0, 1)$ independently from a standard normal and returning $Z = X/\sqrt{Y_1^2 + Y_2^2 + Y_3^2}$. This distribution has $\mathbb{E}[Z] = 0$ and $\mathbb{E}\big[Z^2\big] = 3$, and its density is $\mu(z) \propto 1/(1 + z^2)^2$.

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
