[Reviews · NeurIPS 2019]

Reviewer 1



Overall: I like the paper -- node differential privacy has been shown to be extremely challenging to achieve -- and consequently, this work is a solid algorithmic advance. I think it deserves publication at neurips. That being said, there are two aspects of the paper that render it less useful than I would have liked it to be. The first is that almost no intuition is provided, which makes the algorithm rather opaque. Why is the "concentrated degrees" property needed? Where does the analysis break when this property is not there? A detailed discussion of this in my opinion is necessary. The second is that no constants are provided and all smoothed sensitivity calculations are carried out with a O notation. For example in Lemma 3.1 the local sensitivity calculation is done with a O. While this is fine for a theoretical paper, in my opinion this considerably lowers the value of this work. Any practitioner who would like to implement the algorithm would have to work out all the hairy details from scratch; this is exacerbated by the fact that these exact numbers are indeed needed to implement the algorithm correctly. In my view, addressing these two aspects would make the paper considerably stronger.

Reviewer 2



Positives about the paper: it states a clear, concrete problem, and provides a clear solution. It is (from a mathematical standpoint) nicely written. It is particularly nice that lower bounds are given as well as upper bounds. I enjoyed reading the paper. Negatives about the paper: it is not at all clear to me what the motivation for the paper is, other than people have looked at variations of the problem in the past, and differential privacy for random graphs is intrinsically interesting to a (small) set of mathematically inclined people. That is, it's not clear why anyone needs node differential privacy for Erdos-Renyi graphs -- either for an application, or even for other possbily related mathematical problems. In particular, while the paper clearly demonstrates the proposed algorithm achieves the formal definition of differential privacy, it's not clear how, for example, simply revealing the true parameter p affects the privacy of any individual node, particularly in a random graph. While the paper is working within a well-established framework, at least for this reader, a lack of explanation of this point (why one wouldn't just return p if known or the actual edge density) made it more difficult to understand the motivation for this type of result. The lower bound only holds for constrained graphs, not specifically for random graphs. I would not mind seeing the paper accepted; in general, I support mathematically interesting papers. However, it's hard to make a strongly compelling based on probable limited interest to a NIPS audience. But it should certainly be published somewhere, and NIPS is perhaps as reasonable a home as any. Detailed points: Any insight/thoughts on beta at line 140 would be welcome. The statement that it's a parameter to be determined later is a rather oblique. I personally would have like more clarity at lines 227-228; it's written rather vaguely. (I'm not sure what "suitably small" is, or what ie means to not affect the error in a "significant way".) I assume it is a space issue, as the rest of the writing is much more formal; these lines left me worried. ------- The reviewer thanks the author(s) for the detailed response and feedback. It may be useful to make clear some of this motivation in the revised version of the paper, e.g, this may serve as a building block in other protocols, and the connections to Lipschitz extensions. This reviewer found it helpful. Based on the detailed response clarifying issues raised by this reviewer as well as the other reviewers, this reviewer is raising the prior review score from a 5 to a 6. The reviewer also agrees with other reviewers to recommend acceptance.

Reviewer 3



The fact that a node-DP polynomial time algorithm is available and with almost the same error guarantees as the non-DP algorithm is quite an achievement. I haven't checked all the details of the proofs, but the reasoning seems to flow. The lower bound is also interesting, mostly because it shows that novel techniques are needed to get improvements for G(n,p). One doubt that I have is how the value of s (2nd to last line of Alg 1) can be computed efficiently. The paper feels a bit unpolished in the presentation, especially in terms of conveying intuition and explaining what is going to happen next. The pseudocode in particular feels unnecessary and can easily be replaced with a more thorough description in the text. Please use the correct accents above the 'o' of Erd\H{o}s: in LaTeX, use \H{o}, not \"{o}. Please do not italicize "et al.", as per most manuals of style.

[Author Response · NeurIPS 2019]

1 We thank all of the reviewers for their careful reading of our work. In particular, several reviews made helpful
2 suggestions for how we could improve the presentation, which we will implement in the next revision:

   - 3 We will add more intuition for our construction, to address the comment that our presentation is too dense. In
   4 particular, we will discuss how our estimator can be viewed as a natural "graph analogue" of recent estimators
   5 for classical statistics (and discuss why those estimators cannot be applied directly).

   - 6 We will add explicit constants to the description of the algorithm and its analysis. We point out that most of
   7 the constants appear inside the proof of Lemma 3.1, so there is no obstacle to giving explicit constants.

   - 8 We will clarify the efficient computation of the smooth sensitivity bound $S(G)$. In fact, the only reason it
   9 appears difficult to compute is because we neglected to write that $\ell$ can be restricted to integers in the range 0
   10 to $O(1/\beta)$. Given this fact, it is trivial to compute $S(G)$ efficiently as the maximum of $O(1/\beta)$ numbers. Even
   11 better, by analyzing the derivative (see lines 186–189), it suffices to consider the max of only three numbers.

   - 12 We will give intuition for why the concentrated-degrees property is necessary, as requested by one of the
   13 reviewers. In particular, if we allow the graph to be arbitrary, then the sensitivity of the number of edges is
   14 high, which precludes accurate differentially private estimation. That is, we cannot privately distinguish the
   15 empty graph from a graph where a single node has degree $n - 1$.

   - 16 We will fix all typographical and style errors and clarify all technical points raised by the reviewers.

17 One reviewer also raised questions about the motivation for our work, characterizing it as "esoteric." Our work is about
18 the algorithmic foundations of privacy. Last year's NeurIPS had numerous papers on the foundations of differential
19 privacy (13 last year, by our count), including at least one oral and one spotlight presentation, suggesting a robust
20 interest in this direction within the community. In the revision we will highlight the compelling motivation for our work,
21 which addresses perhaps the most basic possible question about differential privacy for graphs—privately counting the
22 number of edges in the graph to high accuracy.

23 Edge density estimation is a very simple and natural question that is a special case or subroutine of essentially any
24 statistical estimation problem involving graphs. However, it is easy to show that this problem requires large error on
25 worst-case graphs, so most work has focused on how to give more accurate algorithms for realistic graphs. Degree-
26 concentrated graphs are one of the most basic cases where one could hope to give more accurate answers, and even that
27 was an open question prior to our work. So our results are not limited to those interested in the details of random graph
28 models. Beyond the problem itself, the technical heart of our work is an efficient low-sensitivity estimator, sometimes
29 called an efficient *Lipschitz extension*, for degree-concentrated graphs. Efficient Lipschitz extensions are at the heart
30 of many problems in differential privacy, including both iid settings and graph settings. So we believe our results are
31 broadly interesting to those working on the algorithmic aspects of differential privacy.

32 The reviewer also raised a valid question of why differential privacy is necessary for the specific problem of estimating
33 the fraction of edges in a random graph. If we are understanding correctly, the reviewer is asking why there is any real
34 privacy risk for releasing the number of edges in the graph.[1] We point out that one could raise a similar question about
35 one of the best-studied problems in privacy, which is privately estimating a sum of numbers. There one can also make
36 intuitive, or even formal claims that releasing a *single* sum is unlikely to harm individual privacy. However, there is
37 now a large body of work (starting with the celebrated work of Dinur and Nissim, PODS'03) showing how releasing
38 *multiple* simple statistics, including sums or the number of edges in a graph, can lead to spectacular privacy violations.
39 This literature is too extensive to survey here, but we can give some examples:

   - 40 A system for analyzing graph data may allow computing the number of edges in different subgraphs. If one
   41 releases the exact number of edges in multiple subgraphs that differ only on your node, then the difference
   42 between these two numbers is the degree of your node. So simply releasing the number of edges gives no
   43 privacy if we can obtain this statistic for related subgraphs.

   - 44 More generally, releasing the number of edges in many subgraphs (even ones that are not as carefully chosen)
   45 will allow reconstruction of the entire graph. Reconstruction of the graph would enable various significant
   46 attacks (e.g. the seminal work of Backstrom, Dwork, and Kleinberg, WWW'07).

47 More generally, algorithms for computing simple statistics will not be used in isolation, but instead will be used as part
48 of a larger system, and thus it is important to develop differentially private algorithms for simple statistics, even when it
49 is not immediately obvious that these statistics can lead to privacy breaches in isolation. Differential privacy is the only
50 framework we know of that allows for arbitrary *composition* and consequently enables modular algorithmic design.

we want to estimate. Also, our results apply to degree-concentrated graphs for which there is no single parameter $p$ to release.

## Footnotes

[1]The reviewer also asks why we can't simply release the parameter $p$, but this solution isn't valid since $p$ is precisely the parameter


[Meta-Review · NeurIPS 2019]

This paper presents a method for estimating the parameter of an Erdos-Renyi random graph model from a sample graph under node differential privacy. The paper contributes a novel algorithmic technique to achieve node DP via the smooth sensitivity framework which might be useful for other related problems. When preparing the final version of the paper the authors must address the presentation issues raised in the reviews, and in particular make the constants in the algorithms explicit to ensure the method can be implemented by practitioners.